

# Influence of measurement errors on the results of the Brutsaert–Nieber analysis of flow recession curves

Jacek Kurnatowski

Department of Hydroengineering, West Pomeranian University of Technology in Szczecin, Al. Piastów 50, 70-311 Szczecin,
Poland

*Correspondence to*: Jacek Kurnatowski (jkurnatowski@zut.edu.pl)

**Abstract**. The classic analysis of river flow recession hydrographs $Q(t)$ developed by Brutsaert and Nieber (1977) leads to
the graphical representation of the relation $ln(-\mathrm{d}Q/\mathrm{d}t) = ln[f(Q)]$ in a form of scattered points cloud. The paper presents
the analysis of measurement errors of river flow values as one of possible factors generating this dispersion. The relevant
numerical experiment shows the high similarity between the set of points obtained as a classic analysis result and the
experiment output. It has been proved that constant time steps between consecutive flow measurements subject to random
errors generate a systematic error of the analysis results and should be replaced by intervals of variable duration, resulting
from the condition of constant stage/depth decrements.

**Keywords**: recession curves, Brutsaert–Nieber analysis, measurement errors, bias

## 1 Introduction – historical review

Over a century has passed since the publication by Joseph Boussinesq the equations describing the unsteady flow in
a saturated zone of a porous medium (Boussinesq, 1904); nevertheless, the problems concerning the set of solutions to these
equations and practical applications for them still remain unresolved. This results on the one hand from the variety of the
20 solutions, on the other – from the complexity of the real situations being modeled by these equations. Both factors cause
growing diversity of the research trends: several works focus on the analytical or numerical aspects of the solutions, using
various methods and formulating newer solutions and algorithms concerning subsequent boundary conditions, while some of
them (hereinafter referred to as conceptual ones) deal with unifying and simplifying solutions, possible to apply in
a relatively wide range of conditions, in particular having a form of lumped storage models (Tallaksen, 1995).

The lumped storage models in most cases consist in presenting the aquifer as a single nonlinear reservoir which includes
also the linear relation as a special case, i.e.:

$$Q = \alpha S^{\beta} \tag{1}$$

where $Q$ is a river flow [m³s⁻¹] and S is the conceptual reservoir volume [m³]. After considering the mass balance condition
in an unsteady flow and assuming no additional losses and recharging processes, Eq. 1 yields the initial value problem:

$$\frac{dQ}{dt} + aQ^b = 0, \quad Q(0) = Q_0 \tag{2}$$

where:

$$a = \alpha^{\frac{1}{\beta}}\beta; \quad b = \frac{2\beta - 1}{\beta} \tag{3}$$

The initial problem (2) has a solution:

$$Q = Q_0 e^{-at} \quad \text{at } b = \beta = 1 \text{ (linear reservoir),} \tag{4}$$





$$Q = \left[ Q_0^{1-b} - a(1-b)t \right]^{\frac{1}{1-b}} \quad \text{at } b \neq 1. \tag{5}$$

In particular, for $\beta = 2$, $b = 1.5$ (quadratic reservoir):

$$Q = \frac{Q_0}{\left( 1 + \frac{a\sqrt{Q_0}}{2}t \right)^2} \tag{6}$$

In the middle of the twentieth century the problem of recession seemed to be solved. Werner and Sundquist (1951)
presented the recession curves as a sum of three exponential functions with different coefficients; Toebes with his
collaborators defined a set of possible equations of recession curves including the simple exponential, double exponential
and hyperbolic (Toebes and Strang, 1964; Toebes et al., 1969) formulating conditions under which a certain type of curve
can be expected. However, the publication of Brutsaert and Nieber (1977), in which the method of recession curves analysis
was proposed, turned out to be a breakthrough having a significant impact on future research. The essence of this method
was the analysis of the relation $\ln|dQ/dt| = \ln[f(Q)]$. Since this relation, developed on the basis of flow records during the
recession phase, usually creates the set of dispersed points, authors recommended accepting the lower envelope of this set as
a basis for determining the Master Recession Curve. The slope of the envelope line, i.e. the exponent $b$ from Eq. (2) can be
different at initial and later phases of the recession. Brutsaert and Nieber were of the following opinion: „*This procedure
avoids the uncertainty regarding a proper time reference after each rainfall event, and it eliminates the effects of
evapotranspiration*". It follows that the authors perceived the reasons of the set points dispersion in an impact of some
disturbing factors, like evapotranspiration, as well as overland flow and quick subsurface flow (Rupp and Selker, 2006 b).
Fig. 1 shows the typical example of this analysis performed for the set of recession curves recorded in the years 1939-2011 at
the Lochsa River, Idaho. The slope of the averaged recession line was calculated using the organic correlation fitting method
(LOC) which is an alternative method to the commonly used ordinary least squares procedure (OLS). Since LOC provides a
unique line identical regardless of which variable is used as a response one, it is recommended in cases where the primary
interest is in the line slope (Helsel and Hirsch, 2002).

The method developed by Brutsaert and Nieber (almost 40 years later explicated by Chor and Dias, 2015), hereinafter
referred to as BN77, has gained a considerable popularity and is still in use by many researches (e.g. Szilagyi et al., 1998;
Szilagyi and Parlange, 1998; Rupp and Selker, 2006 (a), (b); Sánchez-Murillo et al., 2014; Ghosh, 2015). Admittedly,
Anderson and Burt (1980) warned against graphical techniques of interpreting recession curves that could lead to false
results, but the BN77 method was (and, it may be believed, still has) a large impact on the research techniques of the later
period characterized by the intensification of the controversies about the problem. Tallaksen (1995) states that simple linear
model does not satisfactory represent the recession curve over a wide range of flows. Wittenberg (1999) is of a similar
opinion claiming that more realistic alternative to linear reservoir function is to assume $b = 2$. Supporters of the model's
nonlinearity are also Aksoy and Wittenberg (2011) and Ali et al. (2012). De Rooij (2014) argues that the linearity of the
relation is possible only at coincidence of favorable circumstances, including constant thickness of an aquifer and negligibly
small slopes. On the other hand, Chapman (1999) claims that the storage–discharge relation may vary from linear (for a
confined aquifer) to quadratic (for an unconfined flow), which, due to the unavoidable physical limitations of aquifers,
suggests the advantage of a linear form over a nonlinear one. This position is supported by Fenicia et al. (2006) claiming that
nonlinearity, if it exists, is an apparent phenomenon resulting from an error caused by groundwater recharge. Kleidon and
Savenije (2017) use variational principles to show the correctness of the linear model. Some studies tested the statistical
hypothesis of the model's linearity (e.g. Thomas et al., 2015) or analyzed the reasons for the controversy, such as the paper of
Dralle et al. (2017) showing that the results of identification of recession parameters depend on the adopted methodology
(and are therefore subjective). Another group of the studies omitted the linearity–nonlinearity problem and focused on the



relationships between the catchment features and the character of the recession curves, e.g. Raaia (1995), Vogel and Kroll (1996), Krakauer and Temimi (2011), Lanni et al. (2011). In light of the diversity of approaches, the problem of recession curves should not be considered to be resolved definitely.

The presented work is a contribution to this long–drawn discussion, aimed towards at least partial explanation of the
problems arising during the performance of the recession curves analysis.

## 2 The origin of the analyzed problem

The identification and calibration of the conceptual model of the recession can be carried out in two ways. The first one requires the solution of the initial problem determined on a basis of Eq. (2) and then the optimization of the parameters of this solution at the target function related to one of the well–known criteria (e.g. Nash–Sutcliffe). The second way is the
BN77 method which can be performed directly on the basis of raw measurement data and does not require preliminary mathematical assumptions of recession characteristics. Thus, the parameters of Eq. (2) can be obtained as a result of a graphical interpretation of the relation $\ln|-dQ/dt| = f[\ln(Q)]$. Nevertheless, the flow values, generally obtained from the flow curves and requiring measurements of water stages, are subject to an error resulting from a measurement accuracy. The error of flow measurement affects in turn the error of determining its gradient, and the more strongly, the more often the
observations are made and the shorter is the period $\Delta t$ between consecutive measurements.

As it seems, the impact of such errors on the results of the BN77 analysis has not been sufficiently investigated so far. Although Lamb and Beven (1997) mentioned measurement errors as a component of the total error of the analysis, Rupp and Selker (2006 a) criticized the use of fixed intervals $\Delta t$, especially at high flow variability, there is no estimation of these errors influence on the BN77 results.
The estimation of the errors impact on the BN77 results has been presented below. The analysis contains theoretical considerations visualized by relevant numerical experiments results.

## 3 Analysis of numerical bias in the BN77 method

The numerical bias occurrence in the BN77 method is the effect of replacing the flow derivative with respect to time $dQ/dt$ by a quotient of finite differences $\Delta Q/\Delta t$ and depends on the value of the sum of truncated terms in the Taylor expansion of
the $Q(t)$ function. Thus, this bias should be considered separately for every type of the recession curve as follows:

- The linear reservoir: Determining the relative value of the bias as:

$$\Theta = \frac{\frac{dQ}{dt} - \frac{\Delta Q}{\Delta t}}{\frac{dQ}{dt}} \tag{7}$$

the bias for the linear reservoir yields:

$$\Theta = 1 + \frac{e^{-c} - 1}{c \cdot e^{-\frac{c}{2}}} \tag{8}$$

where $c = a \cdot \Delta t$. The direct assessment of the bias magnitude according to Eq. (8) requires an earlier analysis of the parameter $c$ variability, which can be troublesome considering theoretically large range of both factors included in this parameter. Therefore, from the practical point of view better is to analyze the frequency of stages/flows measurements assuming the given time period, e.g. determining the number of measurements (intervals $\Delta t$) $m$ during the period $t_1$ of reducing the flow by half, as follows:



$$Q_0 e^{-at_1} = \frac{Q_0}{2}; \ t_1 = \frac{ln\,2}{a} = m \cdot \Delta t; \ c = \frac{ln2}{m} \tag{9}$$

Thus, Eq. (8) can be rewritten as:

$$\Theta = 1 + m \frac{2^{-\frac{1}{m}} - 1}{ln2 \cdot 2^{-\frac{1}{2m}}} \tag{10}$$

The graph of the relation (10) is presented in Fig. 2.

5   • The quadratic reservoir: Applying the same method as above the relative numerical bias equals to:

$$\Theta = 1 + \frac{\frac{\left[1 + c_1\left(i + \frac{1}{2}\right)\right]^3}{[1 + c_1(i+1)]^2} - \frac{\left[1 + c_1\left(i + \frac{1}{2}\right)\right]^3}{[1 + c_1 i]^2}}{2c_1} \tag{11}$$

where $c_1 = k \cdot \Delta t, \ k = \frac{a\sqrt{Q_0}}{2}, \ i = \frac{t}{\Delta t}$.

Opposite to the linear reservoir case the relative numerical bias for the quadratic one varies in time; however, the analysis of the bias range can be carried on similarly to the linear reservoir one, i.e.:

10   $$\frac{Q_0}{(1+kt_1)^2} = \frac{Q_0}{2}; \ t_1 = \frac{\sqrt{2}-1}{k} = m \cdot \Delta t; \ c_1 = \frac{\sqrt{2}-1}{m} \tag{12}$$

The graph of the relation (11) for consecutive time steps and different measurements number $m$ is presented in Fig. 3.

The analysis of Figures (2) and (3) clearly proves the lack of significant influence of the numerical bias on the BN77 results, except the situation where measurements of stages/flows are carried out relatively rarely. Since such a situation rather does not happen in practice (if happens, the results are unlikely to be accepted as a field data source for any data 15   processing), the influence of the numerical bias can be neglected in further considerations and the flow derivatives replaced by the relevant finite differences quotients without losing the accuracy of the analysis.

**4 Analysis of measurement random errors in the BN77 method**

The flow values as well as their derivatives with respect to time, calculated as relevant quotients of finite differences, are raw field data records subject to BN77 analysis. These values are usually not obtained from direct measurements but calculated 20   on the basis of predetermined flow–stage relations (the rating curves) by the stage surveying. In each case the flow value is subject to an error which is a superposition of the rating curve error and the random error of a stage measurement. The influence of the stage measurement error on the BN77 results is analyzed below.

Let the general form of the rating curve be given as $Q = Q(H)$, where $H$ [m] is a river depth at a gauging station; then the flow derivative with respect to time can be expressed as:

25   $$\frac{dQ}{dt} = \frac{\partial Q}{\partial H} \cdot \frac{dH}{dt} \tag{13}$$

Introducing the stage/depth measurement random error $\varepsilon$ the rating curve yields the flow value $Q_1$ which is subject to some error in relation to the real value $Q$, i.e.:

$$H_1 = H + \varepsilon; \ Q_1 = Q(H_1); \ \frac{dQ_1}{dt} = \frac{\partial Q_1}{\partial H_1} \cdot \frac{dH_1}{dt} \tag{14}$$

Assuming that the error $\varepsilon$ is negligibly small in relation to the depth, i.e. $H_1 \cong H$ and $\frac{\partial Q_1}{\partial H_1} \cong \frac{\partial Q}{\partial H}$, the following relation is 30   obtained:

$$\frac{dQ_1}{dt} \cong \frac{\partial Q}{\partial H}\left(\frac{dH}{dt} + \frac{d\varepsilon}{dt}\right) = \frac{\frac{dQ}{dt}}{\frac{dH}{dt}}\left(\frac{dH}{dt} + \frac{d\varepsilon}{dt}\right) = \frac{dQ}{dt}\left(1 + \frac{\frac{d\varepsilon}{dt}}{\frac{dH}{dt}}\right) \tag{15}$$





Since the relation $\varepsilon = \varepsilon(t)$ is not a continuous function but the discrete one (the error appears only during stages measurements performance in consecutive time intervals $\Delta t$) and the numerical bias resulting from discretization is negligible, the derivatives of error and depth found in Eq. (15) should be replaced by a relevant finite differences quotients and Eq. (15) yields:

$$\frac{dQ_1}{dt} \cong \frac{dQ}{dt}\left(1 + \frac{\Delta\varepsilon}{\Delta H}\right) \tag{16}$$

or, after adapting to the BN77 pattern:

$$ln\left(-\frac{dQ_1}{dt}\right) \cong ln\left(-\frac{dQ}{dt}\right) + \Psi \tag{17}$$

where $\Psi$ is the error which the flow derivative is subject to in every time step between consecutive measurements:

$$\Psi = ln\left(1 + \frac{\Delta\varepsilon}{\Delta H}\right) \tag{18}$$

It follows that the error of the flow derivative with respect to time depends not only on the stage/depth measurement error, but also on the flow gradient in time. Since the error $\varepsilon$ may be perceived as independent of the recession phase, the greater absolute values of the relative errors of flow derivatives in later recession phase, when the flow is low and its changes are not as intensive as in the earlier one, should be expected. The increase of the stage measurements frequency also provides the larger errors of the flow derivative.

Assuming the symmetrical distribution of the error $\varepsilon$ the expected value of the error $\Psi$ due to the logarithmic function concavity is negative; therefore, the error $\Psi$ may be perceived as a systematic one (bias) and the biased flow derivative is expected to be greater than the real one while the slope of the regression line for the relation (2) increases.

In order to assess and illustrate the random error influence on the results of the regression analysis due to BN77 method a numerical experiment has been carried out. General assumptions for the simulation of the recession were adopted as follows:

• The rating curve is created on the basis of Chézy and Manning formulas at constant water table slope and constant roughness coefficient;

• Riverbed cross–section is rectangular and wide (hydraulic radius $R_h \sim H$).

Therefore, the real flow $Q$ can be determined by the relation:

$$Q = \frac{B\sqrt{S}}{n}H^{5/3} = KH^{5/3} \tag{19}$$

where $B$ is a riverbed width [m], $S$ water table longitudinal slope [–], $n$ roughness coefficient [$m^{-1/3}$s]. Introducing the stage (depth) measurement error $\varepsilon$ the measured flow $Q_1$ yields:

$$Q_1 = K(H + \varepsilon)^{5/3} = K\left[\left(\frac{Q}{K}\right)^{3/5} + \varepsilon\right]^{5/3} \tag{20}$$

The catchment behavior is adopted optionally: either as a linear reservoir or a quadratic one. The parameters of the simulation are shown in Table 1. These parameters ensure the following features:

• adopted $K$ value corresponds to many cases encountered in practice, e.g. to small lowland river with riverbed width $B = 10$ m and slope $S = 0.1$‰, fairly maintained (roughness $n = 0.033$);

• differentiation of parameter $a$ values for both options ensures identical time of initial flow reduction by half.

The random error $\varepsilon$ is entered as a symmetrical one and calculated in two ways: first, assuming the maximum possible value $\varepsilon_{max} = \pm 5$ mm and applying the probability density function as Beta (3,3), shifted to the left to ensure the function

symmetry relative to the point $\varepsilon = 0$; second, according to the normal distribution with a standard deviation $\sigma = 2.5$ mm.

The results of BN77 analysis for the first 100 time steps of each independent simulation of recession curves are shown in Figures 4 – 8. Fig. 4a shows the measured (disturbed by errors) recession curve with the normal distribution of the errors for




both options (the curve for the case of the beta distribution is optically almost indistinguishable). In order to visualize the irregularities of the obtained recession curve better, these irregularities in Fig. 4b were increased three times.

As the errors $\varepsilon$ are generated randomly for each time step, every run of the experiment outputs unique pattern of points in the BN77 coordinate system. Figures 5 – 8 show the results of the BN77 analysis carried on for four situations, i.e. both

catchment behavior options and both error distributions for the same true recession curve. Each figure contains four independent, randomly generated exemplary graphs, selected from the set of patterns obtained as a result of multiple, independent runs of the experiment. The red line in each graph is the result of the estimation by the linear regression performed with OLS method. Since the slope of the regression line is the estimator for the parameter $b$ (Eq. 2), the obtained values of $b$ vary at each run. In Figures 5 – 8 the graphs denoted as a) show incidentally obtained relatively good

compatibility between obtained and true values of $b$, i.e. the low impact of the bias $\psi$ on the recession analysis, graphs d) – the high one. Graphs b) and c) represent intermediate cases.

Despite the fairly regular, smooth shape of the simulated recession curve the BN77 analysis shows a big spread of points and the similarity between Figures 5 – 8 and Fig. 1 is apparent. In particular, sets of scattered points show the higher variance of flow derivative at low flows, which results from decreasing differences between flow values in subsequent

measurements during the late phase of the recession and the related increase of the relative influence of stage measurement error on the flow derivative. The impact of the bias $\Psi$ manifested in increasing value of the exponent $b$ may vary in a wide range; despite the relatively small measurement errors adopted in the experiment the value of $b$ may exceed 1.9 assuming quadratic behavior of the catchment and normal distribution of errors (Fig. 8d), instead the true value $b = 1.5$.

It should be noted that at the constant value of the bias $\Psi$ its impact on the slope of the regression line vanishes and the

bias affects only the vertical shift of the regression line representing the relation $\ln|-dQ/dt| = f[\ln(Q)]$. Thus, to eliminate the impact of the bias $\Psi$ on the parameter $b$ and avoid misinterpretation of the BN77 analysis for any catchment behavior for linearity/nonlinearity problem, the stage/flow measurements should be carried out not at constant time steps, but at constant stage decrements $\Delta H$. Fig. 9 illustrates the exemplary results of the analysis for the same cases as in Figures 5 – 8, carried on with the assumption of constant $\Delta H$ steps and accordingly varying time steps $\Delta t$. The influence of the error $\Psi$ on the value of

$b$ manifests itself in the insignificant deviations of this parameter from the original values and can be perceived as a random, practically slightly small impact.

## 4 Conclusions

The presented analysis is not a proof of the stage measurement errors as the one and only reason for the scattering of points in the BN77 analysis, of course, but it proves the necessity of taking these errors into account each time when interpreting

the results of the analysis. All hydrological interpretations of the variability of flow derivatives (such as the effect of evapotranspiration) should therefore be carried out only after the elimination of the bias resulting from stage/flow measurement errors. In particular, the records of flows applied as raw data sets for BN77 analysis should be pre-processed considering the measurements frequency differentiation in relation to the recession phase (less frequent in a later phase) in order to ensure constant stage decrements between consecutive flow values adopted for the analysis. Considering the BN77

method as a tool for the resolving of the catchment linearity/nonlinearity problem, such a procedure ensures better quality of this tool.





*Competing interests*. The author declares that he has no conflict of interests.

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





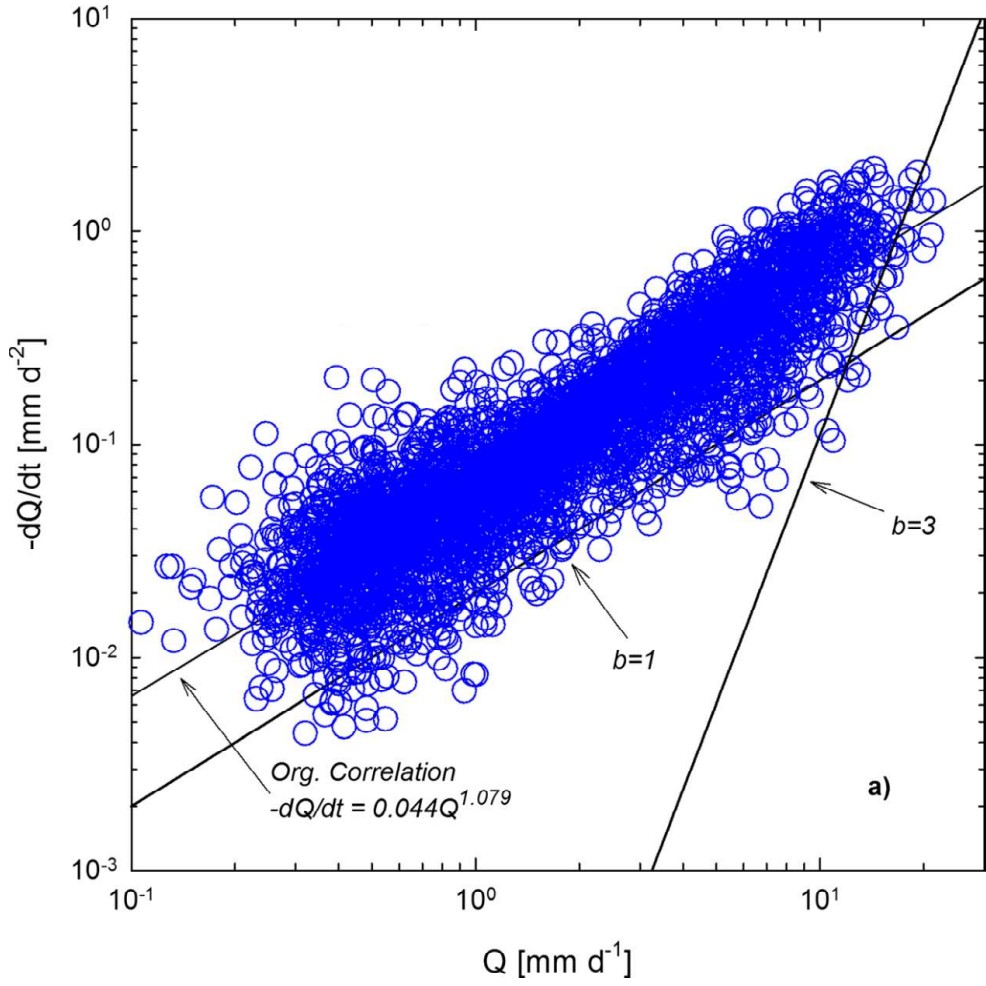

**Fig. 1.** Set of data points showing decline in discharge, $-\mathrm{d}Q/\mathrm{d}t$ [mm d$^{-2}$], versus the average discharge, $Q$ [mm d$^{-1}$], during the period 1939–2011 on Lochsa River, Lowell, Idaho. The drainage area is 3060 km$^2$. The lower envelopes are represented by the lines with slopes $b$=1 and $b$=3. The organic correlation fitting method exhibits a slope of 1.079. Reprint with the permission of Dr. Sánchez-Murillo (Sánchez-Murillo et al., 2014)




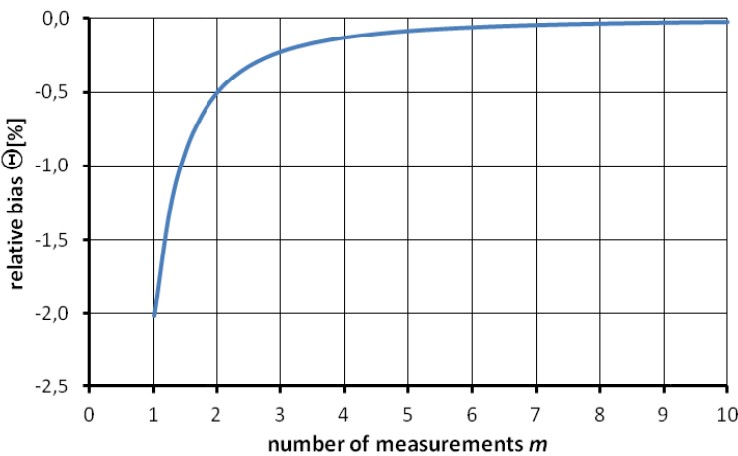

**Fig. 2. The relative bias value Θ versus number of measurements during the time interval of reducing flow by half *m* − linear reservoir model**

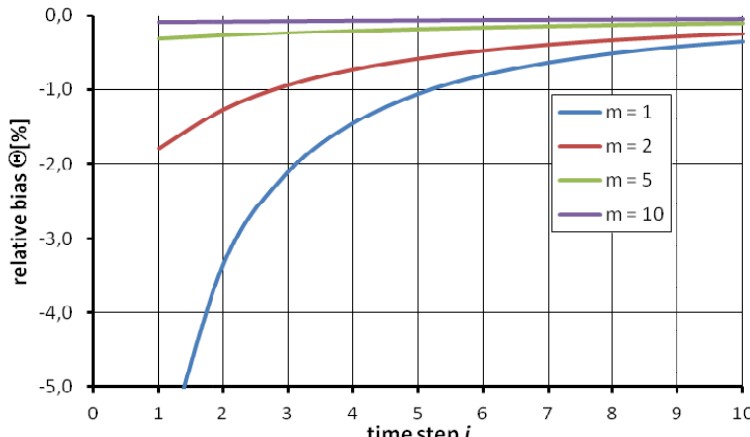

5   **Fig. 3. The relative bias value Θ versus time step *i* and number of measurements during the time interval of reducing flow by half *m* − quadratic reservoir model**





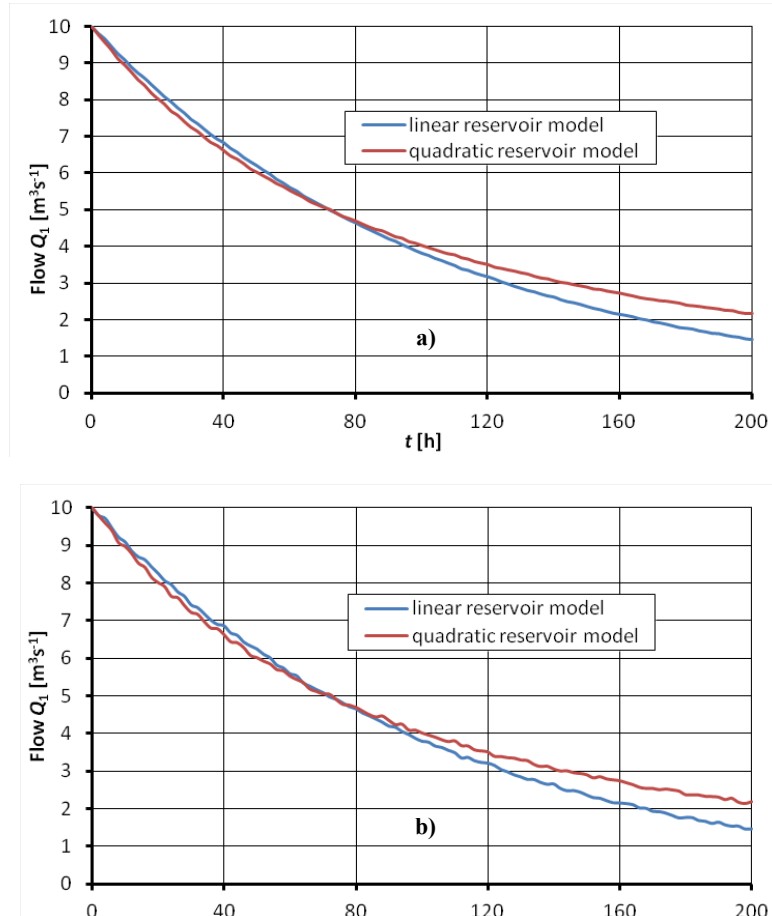

**Fig. 4. Simulated flow recession curve with exemplary stage measurement errors subject to normal distribution: a) errors in the original size; b) errors tripled**





**Fig. 5. The results of BN77 analysis for the simulated recession curve, linear reservoir model, errors subject to the Beta (3,3) distribution, $\varepsilon_{max} = \pm 5$ mm**




**Fig. 6. The results of BN77 analysis for the simulated recession curve, linear reservoir model, errors subject to the normal distribution, σ = 2,5 mm**





**Fig. 7. The results of BN77 analysis for the simulated recession curve, quadratic reservoir model, errors subject to the Beta (3,3) distribution, $\varepsilon_{max} = \pm 5$ mm**



**Fig. 8. The results of BN77 analysis for the simulated recession curve, quadratic reservoir model, errors subject to the normal distribution, σ = 2,5 mm**



**Fig. 9. The exemplary results of BN77 analysis for the simulated recession curve after bias eliminating; a) linear reservoir model, beta distribution; b) linear reservoir model, normal distribution; c) quadratic reservoir model, beta distribution; d) quadratic reservoir model, normal distribution**





**Table 1. Parameters of the recession curves simulation**

| Parameter | Value |
|---|---|
| Initial flow $Q_0$ [m$^3$s$^{-1}$] | 10.0 |
| Hydraulic characteristics $K$ [m$^{4/3}$s$^{-1}$] | 3.0 |
| Coefficient $a$ [h$^{-1}$] – linear reservoir model (Eq. 4) | 0.00963 |
| Coefficient $a$ [m$^{2/3}$s] – quadratic reservoir model (Eq. 6) | 0.00364 |
| Time interval between measurements $\Delta t$ [h] | 2.0 |
| Time of initial flow reduction by half [days] | 3 |