# Peer review of "Influence of measurement errors on the results of the Brutsaert– Nieber analysis of flow recession curves"

_Hydrology and Earth System Sciences, 2018_

## Referee Comment (RC1) · Anonymous Referee #1 · 11 Sep 2018

This paper presents a recession analysis to estimate bucket model parameters under the assumption of error in the rating curve. The paper may provide some interesting results, but it needs serious revisions.

1) The methodology is sometimes difficult to follow. For example, I could not see how the "bias for the linear reservoir yields Equation 8", or what leads to Equation 9, etc. I suggest the steps from one equation to another be made more explicit.

2) The analysis is based on the assumption that the error occurs in measuring the stages, and that such errors are IID with zero mean and constant std. This is a strong assumption that deeds argumentation and subsequent discussion. It should be discussed that errors come from different sources, including e.g. errors in the model inputs, or model structural errors.

3) It is unclear how the fit of Figures 5 to 9 is obtained. If this is obtained by standard least squares, it should be noted that this approach assumes that the errors of ln(-dQ/dt) are iid normally distributed. I am not sure whether this is consistent with the previous assumption of error in the rating curves.

4) Note the previous work of Kirchner, J. W. (2009), Catchments as simple dynamical systems: Catchment characterization, rainfall-runoff modeling, and doing hydrology backward, Water Resour. Res., 45, W02429, doi:10.1029/2008WR006912. This is highly related to the current paper and not discussed. E.g. this paragraph "Brutsaert and Nieber [1977] used plots like Figure 6 to define the lower envelope of dQ/dt as a function of Q, under the assumption that these points would be least affected by evapotranspiration, but in practice, much of the spread in dQ/dt at any particular value of Q may be due to stochastic variability and measurement noise [Rupp and Selker, 2006a], particularly over the short intervals between individual hourly measurements."

5) In terms of structure, the introduction must be more focused (e.g. it is unclear how the presentation of linear vs nonlinear debate is related to the paper; objectives must be better specified), there should be a separation between results and discussion, and the conclusion needs to report relevant findings rather than elements of discussion.

---

## Short Comment (SC1) · 12 Sep 2018

Please note that

Roques, C., D.E. Rupp, and J.S. Selker. Improved streamflow recession parameter estimation with attention to calculation of -dQ/dt. Ad. Wat. Resour., 108:29-43, doi.org/10.1016/j.advwatres.2017.07.013, 2017.

and

Roques, C., Rupp, D. E., Jachens, E., & Selker, J. S. (2018). Comment on "Base flow recession from unsaturated‐saturated porous media considering lateral unsaturated discharge and aquifer compressibility" by Liang, X., H. Zhan, Y.‐K. Zhang, and K. Schilling (2017). Water Resources Research, 54, 3217–3219. https://doi.org/10.1002/2017WR022085

address many of the issues discussed in this paper, and thus these advancements really need to be part of this contribution. Frankly, the objectives of the paper which reads as "it seems like more research could be useful" is far from compelling. Please bring the paper up-to-date, and define a specific and achievable objective for the work. As it stands, this should not be published.
* * *

---

## Author Comment (AC1) · 19 Sep 2018

Dear Professor Selker,

Thank you very much for commenting on my work that you have sent. Indeed, in the analysis of literature I omitted several works, including both papers from 2017 and 2018, which you are the co-author of. This is a serious mistake and I cannot (though with some regret) disagree with your final proposal regarding the publication of my work in its current form. In the revised version I will certainly include these works and refer to them. Nevertheless, I have to mention that the procedure for lengthening the time step proposed by me substantially differs from the use of an exponential function presented

by you, although final results may be similar in both cases. I hope that reading the second version of my work (not completed as yet, I am waiting for the second review) will be a bit more pleasing and you find it valuable.

Sincerely,

Jacek Kurnatowski
* * *

---

## Author Comment (AC2) · 24 Sep 2018

Dear Referee,

Thank you for the review. I refer to particular issues raised by you as follows:

1. "The methodology is sometimes difficult to follow...". Well, I guess this is the result of the imperfections of a good part of authors which follow their own thoughts and think that everyone should keep up with them. In terms of gaps between particular equations – e.g. Eq. 8 is created by substitution of Eq. 4 to Eq. 7 etc, but I understand that these transformations require more precise explanations. While working with the next version

of the paper I shall take care of that.

2. "The analysis is based on the assumption that the error occurs in measuring the stages and that such errors are IID with zero mean and constant std. This is a strong assumption that deeds argumentation and subsequent discussion. It should be discussed that errors come from different sources, including e.g. errors in the model inputs, or model structural errors". In my opinion this is the simplest assumption which can be made in relation to the measurement errors. Any other assumption, e.g. concerning asymmetry of errors, would be risky and, as you have stated, require justification. A separate problem is, however, the PDF for this error. The normal distribution is commonly used as a standard pattern here so I decided to use it as well although I am not convinced about the full correctness of this. The error of river stages measurements may result from many factors, including even such trivial reasons like cleanliness of a gauge rod etc., but the main reason is waving due to wind, in particular when measurements are carried on in the simplest way, i.e. by watching the water level position at a rod. On the other hand the amplitude of waving during the measurement observation is limited and considering the error beyond this limit not necessarily has to be justified. This was the reason of my assumption about the beta (bounded domain) distribution of errors as an alternative PDF. In terms of different sources of errors – considering raw data only the errors may appear as a result of additional processes which have not been considered in an analysis. I am fully aware of that so I do not state that the scattering of points in BN77 results only from measurements, but this kind of errors should be analyzed first. The errors quoted by you are the ones related to modeling procedures and not to measurements, so cannot be discussed while analyzing errors affecting raw data sets.

3. "It is unclear how the fit of Figures 5 to 9 is obtained...". Yes, you are right, I should present the algorithm in a form of consecutive steps and I shall do it.

4. "Note the previous work of Kirchner, J.W. (2009)...". Indeed, during the literature analysis I did not consider some important papers including the Kirchner's one. These

works are important to the point that I decided to rework the introduction and insert a separate discussion of papers dealing with the influence of errors on the analysis results. I hope that the final product will be deprived of the existing shortcomings in this respect.

5. "In terms of structure, the introduction must be more focused...". Well, I shall consider this. I will probably have to give up a part of the introduction and change the rest, albeit with regret.

Anyway, thank you very much again.

Sincerely,

Jacek Kurnatowski

---

## Referee Comment (RC2) · Anonymous Referee #2 · 15 Oct 2018

Review of "Influence of measurement errors on the results of the Brutsaert-Nieber analysis of flow recession curves" by Jacek Kurnatowski.

General:

1. This paper investigates possible errors of analysis when applying this classic method. The errors considered were bias due to numerical approximation of the $\frac{dQ}{dt}$ variable and the bias possibly caused by random errors in flow measurements.

2. First the conclusion is given that the numerical bias can be neglected and that the derivative $\frac{dQ}{dt}$ can be approximated by appropriate finite difference equations. It seems that the detailed analysis given by Thomas et al. (2015, cited in the manuscript) tested various approximations of different orders so I am not sure whether the result shown in this paper is new. While the author cited the Thomas et al. (2015) paper, the citation was not with respect to the various finite difference approximations analyzed in that cited paper.

3. The second error was analyzed by producing a recession curve containing random errors, where the errors were drawn from a probability distribution having a certain variance. The standard finite difference analysis using these data errors produced the cloud of points commonly observed in a $\log(\frac{dQ}{dt})$ vs $\log$ (Q) plot. This result leads to uncertainty in the parameters derived from the model, especially so for the nonlinear reservoir model. The author's approach to eliminate the bias is to suggest using a fixed increment in stage (I would suppose the same would be the case for a fixed increment in discharge), and therefore a variable time step for the derivation of the $\log(\frac{dQ}{dt})$ vs $\log$ (Q) plot. The author proves that this approach does then yield correct model parameters after the approach is used to remove the bias. I question the novelty of this result however. The similar type of approach was proposed by Rupp and Selker (2006a, cited in the manuscript). In the Rupp and Selker approach the time step was scales to make the discharge difference be larger than the precision of discharge measurement. The author does mention that Rupp and Selker criticized the use of fixed time steps for calculating the derivative, and said they did not show the effect of the fixed time step approach on the resulting analysis (BN77). I do not find that to be the case at all with the Rupp and Selker analysis. They not only showed the effect of the fixed time step, but also showed how to fix the problem with the scaled time step.

4. The manuscript has a lot of grammar and sentence structure issues.

Specific.

1. I tried several times to derive equation (8), but was unsuccessful. Each time I come out with $\theta = 1 + \frac{(e^{-c}-1)}{c}$

2. Lines 15-18 on page 5 regarding the error ($\psi$) is not clear after a couple of readings. You probably need to put in more text for the explanation.

3. Explicitly state the error analysis scope in advance. The discussion is limited mainly to the aquifer behavior parameter without regarding the drainage timescale, which is the intercept

of the log plot of the streamflow derivative scattered point cloud. Although the shift of the drainage timescale is mentioned in the nonlinear aquifer simulation in the random error analysis, it's not detailed enough overall.

4. The conclusion that a constant stage increment is necessary for the measurement lacks support for the drainage timescale parameter. Although the slope of the log plot of $\frac{\Delta Q}{\Delta t}$ is consistent after taking this revision and the comparison is straightforward. A shift of the intercept is also observed, the discussion of which has not been given sufficient detail. If the extent of this shift is not acceptable, the measurement suggestion by fixing the stage might be sufficient because the recession analysis does not only focus merely on the aquifer behavior.

---

## Author Comment (AC3) · 19 Oct 2018

Dear Referee,

Thank you for the review. I refer to the matters raised by you in the order in which they appear.

General #1: Yes, this is the paper essence.

General #2: On the basis of a real data set Thomas et al. (2015) analyzed (among other issues) six different finite-difference estimators of the dQ/dt comparing them with the spline algorithm results. I analyzed the single case of centered finite difference

only, but compared the results with the analytical solution, not any other numerical smoothing procedure, so the conclusion is as general as possible and not related to any field data set. Additionally, I presented the results referring them to the frequency of stage/flow measurements in a recession and proved that the simplest scheme of the centered finite difference can be applied practically always without damaging accuracy of the assumption concerning discretization applicability, so both the method and the form of the obtained results presentation are different from those published by Thomas et al. By the way, I needed this conclusion afterwards only to justify the replacement of the analytical derivative by the finite difference quotient whilst moving from Eq. (15) to Eq. (16), nothing more.

General #3: I am not sure whether I understand properly the statement: "...the same would be the case for a fixed increment in discharge". The application of constant increments in discharge leads to the unavoidable spread of the BN77 points – the lower flow values appear in the latest phase of the recession, the bigger is the variance. On the other hand, constant increments in stage result in variable flow increments, due to the nonlinearity of river rating curves. Thus, I see no justification for such a supposition, unless there is an interpretation of this statement which I cannot notice. In terms of the solution proposed by Rupp and Selker – you are right saying that this solution is similar (or at least the search direction is similar), but not identical. Rupp and Selker proposed a method of "variable time step", i.e. increasing time increments along the recession period in order to diminish the impact of measurement noise. This idea was developed later by Roques, Rupp and Selker (2017) as the method of the exponential time step (ETS). I perceive this method as a fundamental progress in the BN77 errors analysis, but still not completed. ETS arbitrarily introduces exponential increase of time increments in a form of the function with the exponent coefficient that is obtained by some pre-processing of the entire recession curve under an assumption of linear behavior of a catchment (there is also additional user-defined coefficient assumed, which limits the time interval). Thus, ETS introduces at least two strong assumptions – linearity of a catchment and exponential increments of the time step, both without any

analytical justification. My solution is, I dare say, much more universal, since it does not impose any restrictions on the catchment behavior nor the function of the time step increase.

General #4: Well, I promise that I will do my very best to polish the final text up and make it free from disabilities and grammar awkwardness.

Specific #1: I apologize; I understand that the transition from Eq. (7) to Eq. (8) may be unclear for a reader. I forgot to mention that the analytical derivative of flow was calculated in the center of the time interval $\Delta t$. The amended version of this text is given in the Supplement.

Specific #2: Of course, I shall try to develop this sentence and make it more clear.

Specific #3 and #4: You are right saying that the intercept value after the bias removal is shifted as well. I did not pay attention to this value since this is (in my opinion) meaningless. The basic parameter indicating the catchment behavior is the exponent "b" and its proper recognition allows to sentence (or saying more gently – to facilitate the conclusion formulation) in a basic and still unresolved dispute over linearity/nonlinearity of catchments. Both parameters, i.e. "a" and "b", are the conceptual model ones and do not have to be physical quantities, just like in the majority of the models of this type. I am of opinion that the intercept should be adopted as it appears after the bias removal, with all its consequences.

Sincerely,

Jacek Kurnatowski

Please also note the supplement to this comment:
https://www.hydrol-earth-syst-sci-discuss.net/hess-2018-413/hess-2018-413-AC3-supplement.pdf

―――――――――――――――――
413, 2018.

**Supplement:**

$$\Theta = \frac{\frac{dQ}{dt} - \frac{\Delta Q}{\Delta t}}{\frac{dQ}{dt}} \tag{7}$$

the bias for the linear reservoir after introducing Eq. (4) and assuming that the derivative of flow is calculated in the center of the time interval $\Delta t$ yields:

$$\Theta = 1 - \frac{\frac{Q_0 e^{-a(t+\Delta t)} - Q_0 e^{-at}}{\Delta t}}{-aQ_0 e^{-a\left(t+\frac{\Delta t}{2}\right)}} = 1 + \frac{e^{-c} - 1}{c \cdot e^{-\frac{c}{2}}} \tag{8}$$

where $c = a \cdot \Delta t$.